# Toxic Metals (Al, Cd, and Pb) in Instant Soups: An Assessment of Dietary Intake

**DOI:** 10.3390/foods11233810

**Published:** 2022-11-26

**Authors:** Daniel Niebla-Canelo, Ángel J. Gutiérrez-Fernández, Carmen Rubio-Armendáriz, Arturo Hardisson, Dailos González-Weller, Soraya Paz-Montelongo

**Affiliations:** 1Department of Toxicology, Universidad de La Laguna, 38071 La Laguna, Tenerife, Canary Islands, Spain; 2Health Inspection and Laboratory Service, Canary Health Service, 38006 Santa Cruz de Tenerife, Tenerife, Canary Islands, Spain

**Keywords:** toxic metals, risk assessment, instant soups, food safety

## Abstract

Instant soups and noodles are one of the most widely consumed commercial food products. These products are made from ingredients of animal (chicken, meat) and/or vegetable origin, in addition to various food additives that prolong the shelf life of the product. It should be noted that instant soups are a dehydrated product, whose water-removal process can increase the accumulation of contaminants, such as toxic metals (Al, Cd, or Pb), that are harmful to the health of consumers. The content of toxic metals (Al, Cd, and Pb) in a total of 130 samples of instant soups of different types (poultry, meat, and vegetables) was determined by ICP-OES (inductively coupled plasma–optical emission spectrometry). The Al content (32.28 ± 19.26), the Cd content (0.027 ± 0.016), and the Pb content (0.12 ± 0.13) in the vegetable soups were worth mentioning. Considering an intake of twenty grams (recommended by the manufacturer), the dietary intake of Al (19.56% of the TWI set at 1 mg/kg bw/week), the intake of Cd (6.59% of the TWI set at 2.5 µg/kg bw/week), and the Pb intake (16.18% of the BMDL set for nephrotoxic effects at 0.63 µg/kg bw/week and 6.84% of the BMDL set for cardiovascular effects at 1.50 µg/kg bw/week) in the population aged 3–10 years, instant soups are not recommended for the population aged 3–10 years, while their consumption does not pose a health risk for adults. However, it is necessary to consider the cooking water used in the preparation of these products, as it may increase exposure to these toxic metals, in addition to the rest of the diet.

## 1. Introduction

The presence of toxic heavy metals (Al, Cd, and Pb) in processed foods, as is the case of instant soups, is currently considered a potential health risk, as they can undergo bioaccumulation processes from the raw materials with which these types of foods are produced. In addition to the above, contamination can occur in the different stages of the production process, as well as migrations of these toxic metals from the packaging to the food in the subsequent storage stages of the product. All of these factors, in addition to the existing trend in the population to consume higher quality products that are healthy and safe, justify the importance of carrying out this type of research.

Instant soups and noodles are among the easy-to-prepare dishes that, due to their low price and great usability, have experienced a significant increase in consumption in recent years. They are industrial products that are presented in packaging containing the dehydrated product and their preparation basically requires cooking in boiling water for a few minutes.

These products contain a number of basic ingredients, such as noodles, as well as animal products (chicken, meat, etc.) and vegetable ingredients, along with a number of food additives, such as sodium glutamate, other salts, etc. These ingredients provide essential minerals and nutrients, but can also contain a variety of contaminants that may pose a health risk. The diet is the main route of access for substances and chemical elements that can cause harmful effects on consumers’ health, such as toxic metals (Al, Cd, and Pb). It should also be noted that this type of product lacks regulatory legislation regarding the level of toxic metals, such as Cd or Pb, in European regulations.

Aluminium (Al) in food has a multitude of uses, from food additives to components of food packaging materials. It is also present in water purification processes [1,2], such as those used in beer and wine production, through zeolites containing aluminium in their composition [3]. However, despite its uses, Al is an element that exerts various toxic effects on health. Several studies have concluded that this element can induce oxidative stress in the cells of the human body [4,5], have genotoxic effects [6,7], and affect the immune system [8,9]. Another of the effects most associated with high Al exposure is its neurotoxicity and its link to Alzheimer’s disease in humans [10,11].

Cadmium (Cd) is a pollutant mainly of anthropogenic origin, due to industrial emissions, the burning of fossil fuels, etc. Although the main route of exposure to Cd is tobacco smoke, food is also an important source of exposure, especially cereals and vegetables, followed by foods of animal origin, such as meat, fish, and seafood, and especially their guts. Cd toxicity particularly impacts the kidneys and bones [12], and has been classified by the International Agency for Research on Cancer (IARC) as a Group I carcinogen [13,14]. Cd food poisoning has been described in a population in Japan to lead to the well-known Itai-Itai disease, which is characterised by multiple fractures, osteoporosis and osteomalacia, kidney damage, emphysema, and anaemia [15,16,17].

Lead (Pb) is a toxic metal and a known environmental pollutant. Its use has been banned in petrol, paints, and water pipes for many years [18]. It is necessary to consider dietary exposure to Pb, as it is an element found in the air, irrigation water, and farmland. Pb affects many systems and apparatuses in the human body, causing neurodevelopmental damage. It is also a mutagenic, carcinogenic, and teratogenic agent [18].

At present, there have been studies on the excessive consumption [19,20] of this type of product and the consequences it has for human health; for example, it is related to the appearance of metabolic syndrome, which is manifested in serious disorders in the health of people, such as obesity, blood pressure problems, diabetes, and cholesterol, among others.

Considering the toxic effects of these metals, as well as the diversity of ingredients in instant soups, which can have both vegetable and animal origins, in addition to the lack of legislation and scientific reports on the content of contaminants in this type of food product, it is necessary to determine the contents of Al, Cd, and Pb in these widely consumed products and to evaluate their dietary intake and the consequent risk to health.

There are also studies, to a lesser extent, that quantified the contents of toxic elements, such as aluminium, cadmium, and lead, in this type of product [21,22,23], and other elements, such as macronutrients including sodium, potassium, calcium, and magnesium [24,25], and their concentrations for dietary evaluations. For all of the above reasons, this research has the following objectives, listed below.

The objectives of this study are: (1) to determine the contents of toxic metals (Al, Cd, and Pb) in different types of instant soups (poultry, meat, and vegetable) and brands, (2) to study the significant differences in the contents of toxic metals according to the types and brands, and (3) to evaluate the dietary intake of toxic metals from the consumption of these products.

## 2. Materials and Methods

### 2.1. Sampling and Sample Treatment

A total of 130 samples of instant soups of different types (poultry, meat, and vegetables) of different brands, marketed in supermarkets and shops, were acquired in Tenerife (Canary Islands, Spain). Table 1 shows the characteristics of the analysed samples.

All of the samples acquired were in a dehydrated, solid state in flow-pack containers, which are characterized by a film with one longitudinal and two transversal seams, forming a perfectly sealed bag containing the product. The inside of this type of packaging is usually coated with aluminium. Nowadays, these types of products, referring to instant soups, are produced by modern and industrial procedures.

The first step was to homogenise the sample and weigh 5–10 g in porcelain capsules (Statlich, Berlin, Germany) (Figure 1). The samples were dried in an oven (Nabertherm, Lilienthal, Germany) at a temperature of 70 ± 10 °C for 24 h [26]. After this, the dried sample was weighed and placed in a muffle furnace (Nabertherm, Lilienthal, Germany) for 48 h following a temperature ramp until it reached 450 ± 25°C for 24 h. Once the greyish–white ashes were obtained [27], they were dissolved in 1.5% HNO_3_ (Sigma Aldrich, Taufkirchen, Germany) to a volume of 25 mL using a volumetric flask and transferred to a decontaminated sampling bottle [28].

### 2.2. Analytical Method

The samples were analysed using an inductively coupled plasma–optical emission spectrometer (ICP-OES), model ICP-OES Thermo Scientific iCAP PRO (Waltham, MA, USA) with an automatic Auto Sampler. The instrumental parameters of the method are shown in Table 2. The instrumental limits of detection and quantification were estimated based on the instrumental response of the instrument. Specifically, they were determined by analysing fifteen blanks under reproducible conditions [29].

### 2.3. Statistical Analysis

Statistical analysis was conducted using GraphPad Prism 8.0.1 software (GraphPad Software Inc., San Diego, CA, USA). The aim of the statistical analysis was to detect the existence of significant differences (*p* < 0.05) in the content of the three toxic metals (Al, Cd, and Pb) between the types of soup (poultry, meat, and vegetable). The Anderson–Darling, D’Agostino, Pearson, and Shapiro–Wilk normality tests were applied [30]. It was found that the data set did not follow a normal distribution, so non-parametric tests were used. The tests used were the Kruskal–Wallis [31] and Mann–Whitney tests [32].

### 2.4. Dietary Intake Calculations

The dietary intake and, consequently, the toxic risk assessment were based on firstly obtaining the estimated daily intake (EDI) (Equation (1)) and then the percentage contribution (Equation (2)) to the tolerable weekly intake values for aluminium (1.0 mg/Kg B. W/Week) and cadmium (2.5 µg/Kg B.W/Week) or the different BMDLs described for lead depending on the toxic effect (cardiovascular effects, 1.5 µg/Kg B.W/Week, and nephrotoxic effects 0.63 µg/Kg B.W/Week) established by the EFSA (European Food Safety Authority) [3,12,18].
EDI = Metal toxic concentration (mg/kg) × daily consumption (kg/day)(1)
Contribution (%) = EDI (mg/day) /Reference value (mg/day) × 100(2)

The calculation of the percentage contributions to the different reference values was performed using the body weights established by the EFSA [33], regardless of the sex of the person, as the difference was not significant. The amount used to obtain the estimated daily intake (EDI) was twenty grams of dehydrated product, which is the portion recommended by the manufacturer for this type of food. It should be noted that this type of product is consumed after cooking with boiling water, with around 250 mL per serving.

It is estimated that 250 mL of water is needed to prepare one serving of this food, as it is the amount recommended by the manufacturer on the product’s packaging; as such, four servings can be extracted from one packet or pot of this type of product, and one litre of water is needed to cook the dehydrated product.

## 3. Results and Discussion

### 3.1. Toxic Metal Levels by Soup Type

The mean concentrations and standard deviations (SD) of the three toxic metals (Al, Cd, and Pb) found in the three types of soup (poultry, meat, and pork), as well as the maximum and minimum values, are shown in the table below (Table 3).

The high standard deviations (SD) found in some of the types of instant soups can be explained by the fact that these are food samples and various factors condition the concentrations of the different metals present. Therefore, there may be great variability in the results, especially when dealing with samples from different manufacturers and, therefore, from raw materials of different origins.

In order to ensure that the quantified concentrations of the three toxic metals (Al, Cd, and Pb) were correct and correctly determined, a recovery study was carried out, the results of which are shown in the following table (Table 4):

In general, the concentrations found for the three toxic metals (Al, Cd, and Pb) were always highest in vegetable soups, followed by poultry and meat soups, because soil is a natural reservoir of contaminants, such as these three metals [34,35].

Regarding Al, the highest mean average concentration was found in soups with vegetable ingredients, with a value of 32.28 mg/kg, reaching a maximum value of 64.69 mg/kg in some of the samples. This was followed by the poultry and meat samples, with mean Al concentrations of 23.53 mg/kg and 14.93 mg/kg, respectively. These results could be explained by the fact that aluminium is an element necessary for the growth of plants, and, therefore, of vegetables. It is found in the soil and promotes the physiological functions of plants, such as the stimulation of root growth, increased nutrient uptake, increased enzyme activity of plants, and increased plant growth [36].

Regarding Cd, again, the samples with the highest concentrations were those of the vegetable type, with a mean average concentration of 0.027 mg/kg, followed by poultry and meat soups; in this case there were no differences between their concentrations, with both types having a value of 0.019 mg/kg. The higher concentrations in the samples of vegetable origin can be explained by the fact that the soil serves as a reservoir of contaminants, either naturally, although to a lesser extent due to human activities, or anthropogenically, as is the case for Cd [37]. In addition to soil, water can also be an important reservoir of Cd, which reaches plants and vegetables through this route [37]. In contrast to aluminium, Cd is not an essential element for plants and vegetables, and is, therefore, considered a toxic element [38].

Finally, in the case of Pb, the same trend as that observed for the other two toxic metals (Al and Cd) was repeated; the highest concentration was found in vegetable soups, with a value of 0.12 mg/kg, followed by poultry soups (0.094 mg/kg) and meat soups (0.059 mg/kg). The explanation again is that the soil where plants and vegetables are grown [39], and the water used to irrigate these crops [40], is a natural and also an anthropogenic reservoir of lead, which means that this element reaches the vegetables from which this type of food is produced.

Once the statistical studies for the different types of soups (poultry, vegetables, and meat) had been carried out, it was found that the results did not follow a normal distribution, so non-parametric tests were applied, specifically the Mann–Whitney U test. The statistical analysis detected significant differences (*p* < 0.05) for Al between the three types of soups, i.e., meat vs. poultry (*p* = 0.0157), poultry vs. vegetables (*p* = 0.0021), and vegetables vs. meat (*p* = 0.0009). The same occurred for Pb, where there were significant differences (*p* < 0.05) for the three types of soup, i.e., meat vs. poultry (*p* = 0.0079), poultry vs. vegetables (*p* = 0.0100), and vegetables vs. meat (*p* = 0.0002). Finally, the statistical analysis showed significant differences (*p* < 0.05) for Cd between two of the three types of soup, i.e., meat vs. poultry (*p* = 0.0179) and poultry vs. vegetable (*p* = 0.0167), but not for vegetable vs. meat (*p* = 0.5959).

The significant differences found were due to various factors, such as the different types of poultry, different feeds used for the animals used to produce the meat for the soups, vegetable soups and soups of animal origin, the contamination of the soils where the vegetables were grown, and many other factors.

As mentioned in the Introduction, one of the main routes of exposure to these toxic metals is through food, which is influenced by many factors, such as pollution. In the case of the three toxic metals that are the focus of this research, these are elements whose origin, apart from being natural, is mostly anthropogenic, due to human activity. This means that these toxic metals reach all environmental spheres, including soil, water, and air, accumulating especially in the soil of the crops from which some of the raw materials used to produce these foods, such as vegetables, are obtained. 

In addition to the contamination of soil and water with which these crops are irrigated, this type of product can be affected by contamination processes in the different stages of the production chain and the migration of substances from the container in which they are stored. It is also necessary to take into account the great variability in raw materials and other ingredients, such as additives and preservatives, used in the production of this type of product, which can contain some of the elements under study here.

As instant soups have not been analysed very often from the point of view of toxic metal contents (Al, Cd, and Pb), there is very little in the literature on this food product, so it was decided to seek similar foods in order to compare the results obtained in the present study with those of other authors (Table 5).

The concentrations of Al found in this study compared with those found by Lopez et al. [22] were similar because they used samples from the same origin (Spain). The results found by Soylak et al. [21] were much lower than those of the present study, which may be due to the origin (Turkey), a non-EU country that is not governed by European food safety legislation.

There were slight differences between the Cd concentrations found in this study and those found in Bangladesh [23], with the concentrations in this study being somewhat higher. This minimal difference may have been due to the origin of the raw materials used to produce this type of food or to the use of food additives permitted in the European Union; even so, these values were below the reference values, so there was no risk to human health.

Regarding Pb, there were notable differences between the concentrations found in this study and those found in samples of Bangladeshi origin by Nazmul et al. [23], with the concentrations of Pb in the samples from Bangladesh being higher than those in the samples from Spain analysed in this study. These differences may have been for the same reason as those observed for Al, as Bangladesh does not follow the sanitary controls on food safety of the European Union. In addition to the above, these samples came from countries with high levels of environmental contamination [41]. The presence of these toxic metals, such as lead, makes it logical that their concentrations were higher in foods (in this case, soups), reaching high values in the blood of these populations [42,43].

### 3.2. Dietary Intake Assessment

The evaluation of dietary intake, as mentioned above, began with the calculation of the estimated daily intake (EDI). For this, a portion of twenty grams of soup in its dry state (not hydrated with 250 mL of water) was taken, as this was the mean average value of the different quantities recommended by the manufacturers of this type of product. The dietary assessment ended with the calculation of the contribution to the EFSA reference values, expressed as a percentage, for Al, Cd, and Pb (Table 6, Table 7 and Table 8).

In general, in the case of the three toxic metals (Al, Cd, and Pb), as the highest concentrations were found in vegetable soups, this type of product presented the highest contribution to the reference value for each of these metals. At this point, it should be borne in mind that these percentages corresponded to the dried product, i.e., they were calculated from a portion of twenty grams of dehydrated product (the portion recommended by the manufacturer), but this type of food is consumed after hydrating it with water, with generally 250 mL of water per portion (portion of water recommended by the manufacturer), so the metal content of the water used for cooking must also be taken into account.

Another aspect to consider is that only the risk from three years of age onwards was assessed, as infants between 0 and 3 years of age are not considered consumers of this type of product because their diet is based on breast milk and other unprocessed foods [44], which does not include instant soups.

In view of the contribution percentages found for Al (Table 6), contributions close to 10% were calculated for three-to-ten-year-olds in the case of meat soups (9.05%) and almost 20% in the case of vegetable soups (19.56%). These contribution percentages showed that there could be a toxicological risk for this age group, because only the dried food was being taken into account, without taking into account the water in which it would be cooked or the rest of the diet. In the case of adolescents, the possibility of toxicological risk from the consumption of this type of food decreases as the percentages were always below 10%, with the exception of vegetable soups consumed by adolescents between ten and fourteen years of age, which reached a percentage of 10.41%. Regarding adults, the percentages were always below 10%, with the highest values being observed for vegetable soups in adults over seventy-five years of age, which could pose a toxicological risk as this part of the population may be prone to consuming more of this type of food.

As the contribution percentages for Cd were evaluated (Table 7) without taking into account the contribution from the cooking water, it can be seen that there was no risk for any age group, as all contributions to the reference values were below 10%. The highest percentages of 6.59% were found in vegetable soups for children between three and ten years old, while the percentages were always lower than 5% for the remainder of the age groups and types of instant soups; therefore, when only considering the contribution of this food, there was no risk regarding Cd, but it should be kept under continuous surveillance.

As previously mentioned, there are two reference values for Pb described by the EFSA [18], one for cardiovascular effects (*) and one for nephrotoxic effects (**). In view of the results found, the highest contributions to the reference values (Table 8) were found in vegetable soups for children between three and ten years of age, exceeding 15% when evaluating the nephrotoxic effects (16.28%); a high value (12.96%) was also found in poultry soups for this age group and this toxic effect. For the remainder of the life stages, from adolescents to older adults, the percentages were always below 10%. Therefore, there could be a risk for children between three and ten years of age consuming any of these types of instant soups.

## 4. Conclusions

Once the evaluation of the dietary intake of toxic heavy metals (Al, Cd, and Pb) had been performed, the highest concentrations of the three metals were found in vegetable soups when consumed by children between the ages of three and ten. There were significant differences for all metals in the three types of soups, with the exception of Cd in vegetable and meat soups. The metal concentrations were found, assuming a portion of twenty grams of dehydrated product, as recommended by the manufacturer, without taking into account the cooking water, which would make its own contribution of toxic metals, other elements, and chemical substances, resulting in significant estimated daily intakes (EDI) of the three metals.

The EDIs were used to calculate the contributions of the three toxic metals to the different reference values, and special mention should be made to the percentage of nearly 20% for aluminium in vegetable soups in children between three and ten years of age and the high percentage for Cd in the same age group. Based on these results, the consumption of instant soups by young children should be low, avoiding excessive consumption, as there could be a health risk in this population. Additionally, only one food was evaluated, not considering the whole diet of the person, and only the dehydrated product was considered, without taking into account the cooking water, meaning that there is a higher probability of toxicological risk when children consume these products.

## Figures and Tables

**Figure 1 foods-11-03810-f001:**
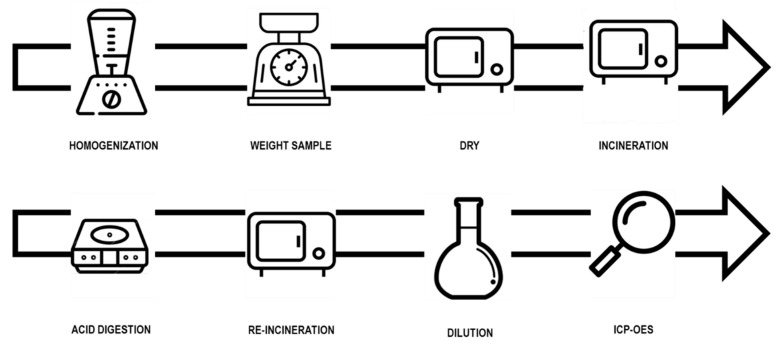
Sample treatment and analysis.

**Table 1 foods-11-03810-t001:** Characteristics of the analysed samples.

Type	No. Samples	Recipe	Container
Poultry	61	Chicken soup, poultry, chicken, chicken, gardener, marigold, and others	Flow-pack
Vegetable	52	Vegetable soup, onion, twelve vegetables, gardener, and others	
Meat	17	Beef Soup, oxtail, veal, Home-style, Stew, Puchero, and others	
Total	130		

**Table 2 foods-11-03810-t002:** Instrumental parameters of the equipment.

Instrumental Parameters			
**RF Power**		**1150 W**	
Nebulizer Gas Flow		0.50 L/min	
Nebulizer Gas Pressure		0.2 L/min	
Auxiliary Gas Flow		0.50 L/min	
Cool Gas Flow		12.5 L/min	
Pump Speed		45 rpm	
Wavelength (nm)	Detection limit (mg/L)		Quantification limit (mg/L)
Al (167.0)	0.005		0.015
Cd (214.4)	0.0007		0.002
Pb (220.3)	0.0009		0003

**Table 3 foods-11-03810-t003:** Toxic metal levels by soup type.

Type	Parameter	Al (mg/kg)	Cd (mg/kg)	Pb (mg/kg)
Poultry	Mean content ± SD	23.53 ± 15.21	0.019 ± 0.009	0.094 ± 0.16
	Max. value	61.79	0.045	0.86
	Min. value	5.11	0.0086	0.015
	Median	18.45	0.016	0.045
Meat	Mean content	14.93 ± 10.79	0.019 ± 0.0069	0.058 ± 0.069
	Max. value	53.20	0.030	0.32
	Min. value	5.28	0.010	0.025
	Median	13.10	0.018	0.043
Vegetable	Mean content	32.28 ± 19.26	0.027 ± 0.016	0.12 ± 0.13
	Max. value	64.69	0.069	0.85
	Min. value	0.70	0.0087	0.017
	Median	35.83	0.024	0.086

**Table 4 foods-11-03810-t004:** Recovery study results.

Metal	Certified Reference Material (CRM)	C. Obtained (mg/kg)	C. Certified (mg/kg)	Recovery (%)
Al	SRM 1515 Apple Leaves	312.60 ± 6.3	286 ± 9	109.3
Cd	SRM 1573a Tomato Leaves	1.40 ± 0.07	1.52 ± 0.04	92.3
Pb	SRM 1515 Apple Leaves	0.456 ± 0.054	0.470 ± 0.024	94.8

**Table 5 foods-11-03810-t005:** Comparison of the contents of toxic metals (Al, Cd, and Pb) with results obtained by other authors.

Type	Origen	Al (mg/kg)	Cd (mg/kg)	Pb (mg/kg)	Reference
Soup	Turkey	6.86–547.7	-	-	[21]
Chicken noodle	Spain	23.97	-	-	[22]
Chicken and vegetable soup		19.54			
Noodles	Bangladesh	-	0.010	0.170	[23]
Poultry soup	Spain	23.53	0.019	0.094	The present study, 2022
Meat soup		14.93	0.019	0.058	
Vegetable soup		32.28	0.027	0.12	

**Table 6 foods-11-03810-t006:** Estimated daily intake and contribution percentage to the tolerable weekly Al intake.

Aluminium (Al)			EDI (mg/Day)		
			Poultry	Meat	Vegetable
			0.47	0.30	0.65
Life Stage	Age (Years)	Bodyweight (kg)	% TWI		
			Poultry	Meat	Vegetable
Children	3–10	23.1	14.26	9.05	19.56
Adolescents	10–14	43.4	7.59	4.81	10.41
	14–18	61.3	5.37	3.41	7.37
Adults	18–64	73.9	4.46	2.83	6.11
Elderly	65–75	76	4.33	2.75	5.95
	>75	71.2	4.63	2.93	6.35

**Table 7 foods-11-03810-t007:** Estimated daily intake and contribution percentage to the tolerable weekly Cd intake.

Cadmium (Cd)			EDI (mg/Day)		
			Poultry	Meat	Vegetable
			0.38	0.39	0.54
Life Stage	Age (Years)	Bodyweight (kg)	% TWI		
			Poultry	Meat	Vegetable
Children	3–10	23.1	4.58	4.72	6.59
Adolescents	10–14	43.4	2.44	2.51	3.51
	14–18	61.3	1.72	1.78	2.48
Adults	18–64	73.9	1.43	1.47	2.06
Elderly	65–75	76	1.39	1.43	2.00
	>75	71.2	1.48	1.53	2.14

**Table 8 foods-11-03810-t008:** Estimated daily intake and contribution percentage to the Pb BMDL (* Cardiovascular effects, ** Nephrotoxic effects).

Lead (Pb)			EDI (mg/Day)					
			Poultry		Meat		Vegetable	
			1.9		1.2		2.4	
Life Stage	Edad (Years)	Bodyweight (kg)	% TWI					
			Poultry		Meat		Vegetable	
			*	**	*	**	*	**
Children	3–10	23.1	5.44	12.96	3.36	7.99	6.84	16.28
Adoslecents	10–14	43.4	2.90	6.90	1.79	4.26	3.64	8.67
Adoslecentes Adults	14–18	61.3	2.05	4.88	1.27	3.01	2.58	6.14
	18–64	73.9	1.70	4.05	1.05	2.50	2.14	5.09
Elderly	65–75	76	1.65	3.94	1.02	2.43	2.08	4.95
Very elderly life stage	>75	71.2	1.77	4.20	1.09	2.59	2.22	5.28

## Data Availability

Data are contained within the article.

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
