# Peer review of "Toxic Metals (Al, Cd, and Pb) in Instant Soups: An Assessment of Dietary Intake"

_foods, 2022, doi:10.3390/foods11233810_

Round 1

Reviewer 1 Report

My comments are included in the PDF file.

Author Response

Dear reviewer,
In the file added we answered all your questinons and adviceds, thank you very much for your dedication and good advices.

Kind regards

Reviewer 2 Report

This manuscript needs an extensive revision. In particular, the results of the method validation should be presented to support the reliability of the analysis results. Furthermore, more attention should be paid to the editing of the article and the English style.

Author Response

(The authors gave the same response as above.)

Reviewer 3 Report

The study evaluated the concentration of Al, Cd and Pb in 130 soup samples which were collected in Spain. From the data results, the study measured an assessment of dietary intake. The manuscript has not been prepared to an acceptable English language usage. Please very carefully edit the revised manuscript and ensure that it conforms exactly to the Journal's format as set out in the Guide for Authors. I encourage the author to use Language Editing Services of MPDI to improve the completeness of the manuscript. Although the topic of this paper is important, it suffers from a number of shortcomings

1.    Overall, the study assessed the concentration of Al, Cd and Pb in instant soups by ICP-OES. Analytical methods as well as calculation methods in this study are still not more innovative than previous studies.

2.    Introduction: The author should highlight the urgency and practicality of this study. Currently, the author is only presenting a separate list of each metal. The author needs to highlight the question “Why is it important to identify Al, Cd and Pb in instant soups? Why not choose other metals?”

3.    The Sampling part: the author needs to present specific information about instant soups in SI, specifically: sample status, material of the container, traditional or modern procedure?,.... The specific information could help the author that easily detect metal contamination in the production process.

4.    I suggest the inclusion of quality control/quality assurance (QC/QA) as one section in the materials and methods part.

5.    Salt and food additives are major ingredients in instant soups samples. With high concentration of  NaCl in the sample, did the author remove the influence of spectral interference, ion interference in the ionization process due to this compound?

6.    Line 120 – 121: the author needs to provide the reasons why using 250 mL of water. In fact, does a person completely absorb 250 ml of water for a meal?

7.    I recommend that 3.1 and 3.2 should be written together.

8.    In Section 3.1, the concentration of metals in the soups samples with vegetable ingradients has the highest value, and the reason was explained by absorption from the soil. à I think this reason is still not convincing. Could you determine concentration of metal in raw vegetables?

Overall, the research was carried out with the idea of novelty not really outstanding. Although 130 samples were analyzed, the QA/QC information was not found in the experiment. The explanations are also very general, not really convincing. The author should revise carefully the article to response the quality of Foods journal.

Author Response

(The authors gave the same response as above.)

Round 2

Reviewer 1 Report

Dear Authors,

I appreciate your contribution to improving the manuscript. Responsible for my every remark. In most cases the manuscript was corrected as suggested. In other cases, responses have been formulated that I am able to accept.

I have no more comments - they were all included in the 1st round of review.

Reviewer 2 Report

The authors have modified the work and the quality has definitely improved.

About validation, how was the precision calculated?